Prediction of cancer cell line-specific synergistic drug combinations based on multi-omics data

Chen Jiaqi
Han Huirui
Li Lingxu
Chen Zhengxin
Liu Xinying
Li Tianyi
Wang Xuefeng
Wang Qibin
Zhang Ruijie
Feng Dehua
Yu Lei
Li Xia
Wang Limei
Li Bing binglijpn2003@aliyun.com
Li Jin lijin@hainmc.edu.cn
College of Biomedical Information and Engineering, Kidney Disease Research Institute at the Second Affiliated Hospital, Hainan Engineering Research Center for Health Big Data, Hainan Medical University , Haikou, Hainan , China
Gomez Shawn
Electronic publication date: 2025 Feb 25
Publication date: 2025
Volume: 13
Electronic Location ID: e19078
Received 2024 Sep 24; Accepted 2025 Feb 10
Copyright: © 2025 Chen et al.
Copyright year: 2025
Copyright holder: Chen et al.
License: This is an open access article distributed under the terms of the Creative Commons Attribution License, which permits unrestricted use, distribution, reproduction and adaptation in any medium and for any purpose provided that it is properly attributed. For attribution, the original author(s), title, publication source (PeerJ) and either DOI or URL of the article must be cited.
License URL: https://creativecommons.org/licenses/by/4.0/

Keywords: Drug synergy prediction, Multi-omics, Machine learning, Cancer, Drug combination

Funding: Natural Science Foundation of Hainan Province 821QN0894, 621MS041 and 824RC514 National Natural Science Foundation of China 32260155 This work was supported by the Natural Science Foundation of Hainan Province [No. 821QN0894, 621MS041, 824RC514]; National Natural Science Foundation of China [No.32260155]. The funders had no role in study design, data collection and analysis, decision to publish, or preparation of the manuscript.

==============================
Compared to single-drug therapy, combination therapy involves the use of two or more drugs to reduce drug dosage, decrease drug toxicity, and improve treatment efficacy. We developed an extreme gradient boosting (XGBoost)-based drug-drug cell line prediction model (XDDC) to predict synergistic drug combinations. XDDC was based on XGBoost and used one of the largest drug combination datasets, NCI-ALMANAC. In XDDC, drug chemical structures, adverse drug reactions, and target information were selected as drug features; gene expression, methylation, mutations, copy number variations, and RNA interference data were used as cell line features; and pathway information was incorporated to link drug features and cell line features. XDDC improved the interpretability of drug combination features and outperformed other machine learning methods. It achieved an area under the curve (AUC) of 0.966 ± 0.002 and an AUPR of 0.957 ± 0.002 when cross-validated on NCI-ALMANAC data. Different types of omics data were evaluated and compared in the model. Literature and experimental verification confirmed some of our predictions. XDDC could help medical professionals to rapidly screen synergistic drug combinations against specific cancer cell lines.

Introduction

Drug combination therapy is a treatment method that involves administering two or more drugs to cure a disease (Bayat Mokhtari et al., 2017). This method has been shown to be more effective than monotherapy. It helps to reduce drug dosage, minimize side effects, and increase the effectiveness of treatment (Al-Lazikani, Banerji & Workman, 2012; Jia et al., 2009; Ma et al., 2014). Over the past decade, the approach to developing new drugs has shifted from the traditional “one drug-one target-one disease” model to the development of multi-drug combination therapies (Hoelder, Clarke & Workman, 2012; Jia et al., 2009). In addition, researchers are also exploring the possibility of repurposing old drugs for a specific cancer or cell line.

Traditional drug development models focus primarily on the design and development of a single target, without fully considering the complex etiology of diseases such as cancer. These models typically target the action of a drug on well-defined biomarker or pathway, in an attempt to achieve a therapeutic effect through a single molecular intervention (Koutsoukas et al., 2011; McGranahan & Swanton, 2015). However, as a highly heterogeneous disease, the onset and development of cancer involves many interrelated factors, including genetic mutations, aberrant cellular signaling pathways, and changes in the tumor microenvironment (Vasan, Baselga & Hyman, 2019). In these traditional models, the lack of a comprehensive understanding of cancer heterogeneity and diversity leads to limitations of treatment. This is because even relatively successful single-target drugs are susceptible to the genetic diversity of tumors (Lavecchia & Cerchia, 2016). For example, a drug-resistant mutation in a tumor cell can cause the entire tumor to evolve in a short period of time, rendering effective drugs ineffective (McGranahan & Swanton, 2015; O’Neil et al., 2016).

In theory, compared to monotherapy, combination therapies can treat the patients more effectively by acting on multiple drug targets or biological pathways at the same time and with fewer doses (Yadav et al., 2015). However, the specific development of drug combination therapies also faces significant challenges. The main one that needs to be addressed is the identification of successful drug combinations (Ling & Huang, 2020). In drug combination therapy, complex interactions between drugs can occur. Drug combinations can be classified as synergistic, additive, or antagonistic based on the differences between the expected response calculated by the reference model and the observed response (Chou, 2006). Synergy is considered when the combined effect of two drugs is greater than the predicted effect of their individual drug potencies. If the effects of each drug neither reduce nor increase the overall effect of the individual drug, it is said to be additive, indicating no interaction. In contrast to synergy, if the total effect is less than the expected effect, the combination is antagonistic. Synergistic combinations allow better therapeutic effects to be achieved with lower concentrations or fewer doses of drugs (Saputra et al., 2018). Nevertheless, the time and cost required to select synergistic drug combinations from millions of possible drug combinations through clinical trials or high-throughput screening (HTS) alone is prohibitive (Bajorath, 2002; Mathews Griner et al., 2014).

With such a huge space for drug combination screening, advances in computational technology and artificial Intelligence have brought new opportunities for the identification and development of drug combinations. Existing computational methods could be divided into three categories, systems biology similarity-based methods, network-based methods, and machine learning-based methods (Guvenc Paltun, Kaski & Mamitsuka, 2021). Zhao et al. (2011) collected information such as ATC classification codes, drug target interactions, and drug indications of individual drugs from FDA-approved drugs. Based on similarity analysis of the information of single drugs in known approved drug combinations, two drugs with a high degree of similarity are defined as possible drug combinations (Zhao et al., 2011). Li et al. (2020) used molecular networks to predict drug synergistic combinations. It used not only the chemical structure of the drug, but also target protein of the drug and the protein connections in the PPI network, and used the Bayesian network to calculate the drug similarity score to obtain the drug synergy.

Machine learning methods were also the most popular methods used to study drug combinations in cancer. They were favored by most scientific researchers because they could learn the correlation between features and extract valuable information from the data (Fan, Cheng & Li, 2021; Guvenc Paltun, Kaski & Mamitsuka, 2021). Celebi et al. (2019) used drug structure, drug target, known drug co-expression networks, gene expression, mutations and copy number changes as features. They used the average expression values of 53 identified modules as labels in a weighted gene co-expression network analysis. The XGBoost model was then used to derive the final prediction score (Celebi et al., 2019). In a previous work, we conducted a study on how gene essentiality and pathway level scores can improve drug synergistic prediction (Li et al., 2020). To explore the effects of different drug combinations at different doses on different cell lines, Julkunen et al. (2020) developed a ComboFM model based on tensor decomposition using multidirectional interactions between two drugs, cell lines, and dose-response matrices as feature data. ComboFM accurately predicted the previously untested drug combinations formed from individual drugs in the training set in a dose-response matrix (Julkunen et al., 2020).

With the continued development of high-throughput technology and the massive generation of omics data, deep learning technology began to be applied to the predicting drug synergy. Preuer et al.’s (2018) built a feed-forward neural network (DeepSynergy) using chemical descriptors and cancer genomic information to predict anti-cancer drug combinations for multiple cancers by comparing deviations with four theoretical models (Loewe Additivity, Bliss Independence, Highest Single Agent, or Zero Interaction Potency). Pak et al. (2023) combined deep learning with drug target networks to create a new prediction model, NetGP, which significantly improved the prediction efficiency. However, there was still much room for exploration in the application of existing machine learning methods to drug combination therapy.

In this work, we propose a new drug combination prediction model called XDDC, which uses pathway information as a link between drug combinations and cell lines, thereby improving biological interpretation. Impressively, the XDDC model performed well in predicting synergistic drug combinations. To reduce the influence of sparse matrices on the prediction results, data quality was screened and the optimal model was first evaluated using methods such as grid search for optimal parameters and weight settings. In terms of feature generation, eight different feature groups, including drug structure, drug toxicity, drug pathway and cell line multi-omics data, were randomly combined to build the model, and seven of them were selected as the optimal feature combinations based on the model evaluation index. Considering that there were only 38 cell lines in the screened NCI-ALMANAC data, the number of drug combinations with synergistic ability in a single cell line was too small, leading to an imbalance of positive and negative data, we integrated multiple cell lines together to make predictions, and adjusted the score thresholds for determining the synergistic drug combinations downward, and selected various methods to deal with the imbalance of data, such as under-sampling and over-sampling, and fully cross-validated the model to confirm the predictive performance of the model. The cross-validation confirms that the predictive performance of the model is robust and does not suffer from overfitting due to oversampling. The prediction results show that our constructed XDDC model outperforms the remaining traditional machine learning models. Finally, we further predict the classification results on NCI-ALMANAC data (Holbeck et al., 2017) and provide literature validation and experimental verification of the positive prediction results.

Method

Framework of XDDC

The process of XDDC is illustrated in Fig. 1. We employed drug combination-cell line data from the NCI-ALMANAC database and collected relevant drug and cell line feature data from various public databases. To mitigate the sparsity of the input matrix, we conducted data filtering and feature compilation. Subsequently, we utilized five machine learning methods and a deep learning model for modeling, fine-tuned the models and performed evaluations. Finally, predictions were made using the optimal model, and the results were validated.

Figure 1 The framework of the XDDC.

XDDC is primarily divided into three parts. The first part involves data collection, encompassing drug combination-cell line data, drug features, cell line features, and pathway data. The second part is feature compilation, transforming individual drug data into drug combination information, establishing connections between cell lines and genes to represent cell line-pathway relationships and processing drug combination-cell line scores. The third part utilizes various machine learning methods for model construction and subsequent analysis of the filtered data. For detailed information, please refer to the Methods section.

Drug and pathway data screening

We collected the names of 104 drugs from NCI, obtained the target information of drugs from HCDT (Chen et al., 2022), and screened 340 human-related pathways (including genetic data on pathways) from the Kyoto Encyclopedia of Genes and Genomes (KEGG) database (Kanehisa & Goto, 2000). Considering that the sparsity of the matrix will affect the predicted results, we first screened the drug and pathway data according to the following criteria: (1) the drug has at least 10 target genes; (2) there are at least 10 genes in the pathway; and (3) at least half of the drugs in each pathway can act on it. As there was little data about the direct action of drugs on pathway, we assumed that the drug act on a pathway when the target gene of the drug is in the pathway. As a result, we retained 87 drugs and 157 pathways.

Drug features

For drugs, we collected information based on the chemical structure, toxicity, and pathways associated with drug targets. Chemical fingerprinting is the most commonly used structural feature for drugs, allowing the representation of chemical properties at the molecular level (Carracedo-Reboredo et al., 2021). We utilized fingerprints to represent drugs as binary vectors indicating the presence (1) or absence (0) of various chemical features (such as S-S, N-O, and C = CN bonds). We collected the SMILES of drugs from HCDT and PubChem (Kim et al., 2021), generated 166-length MACCS molecular descriptors, and represent each drug as a binary 0–1 vector using the RDkit library (Lovric, Molero & Kern, 2019) in Python. For each drug combination, we used the sum of the two drug vectors as a feature. (i.e., 0 indicates neither drug possesses the structural feature, one indicates only one drug in the combination has the feature, and two indicates both drugs in the combination have the feature). We retained only 109 descriptors as the structural features of drugs after ensuring that each MACCS molecular descriptor was present in at least ten drugs.

We collected toxicity features from the SIDER database, with 16 drugs having no matching results. For these 16 drugs, we gathered information from DrugBank (Wishart et al., 2018) based on an overview of drug toxicity. Keywords were extracted from the descriptions of toxicity in DrugBank and matched to the characteristics in SIDER, capturing toxicities for these drugs. We selected toxicity labeled as LLTs in SIDER and ensured that each side effect had at least one corresponding drug, resulting in the retention of 97 features of drug toxicity. Similarly, each drug was represented as a binary vector, and the feature for each drug combination was represented as the sum of the two drug vectors.

Out of the 87 drugs selected from NCI, a total of 1,571 target genes were obtained from the HCDT database. We filtered out 157 human biological pathways in KEGG, generating 157 features for the subsequent analysis. The individual drug-pathway matrix was denoted as D.

(1) D=[d11⋮dm1d12⋮dm2…⋱…d1,n⋮dmn],di,j={01#(di∩pj)=0#(di∩pj) > 0#.

The drug-pathway matrix D represented whether a drug acted on a particular pathway, where di,j was computed based on the intersection of the target gene set di of drug i and the gene set pjof pathway j. A value of 0 indicated that the drug did not act on the pathway, while a value of 1 indicated that the drug acted on the pathway.

Due to the interactions between pathways, we utilized the Jaccard coefficient to assess the correlation between pathways, and we denoted the pathway-pathway matrix as P:

(2) P=[p11⋮pn1p12⋮pn2…⋱…p1,n⋮pnn],pi,j=#(pi∩pj)#(pi∪pj)#.

The P matrix was an n*n matrix, where pi,j was calculated based on the gene sets of pathway i and pathway j. #(pi∩pj) indicated the number of genes in the intersection of pathway i and j, #(pi∪pj) indicated the number of genes in the union of gene i and j.

Considering that a drug could act on multiple pathways and that there were interactions between pathways, we refined the effect score of a drug on a pathway according to Formula (3):

(3) D~=DP=[d~11⋮d~m1d~12⋮d~m2…⋱…d~1n⋮d~mn],d~ij=D→iT⋅P→j=∑k=1n⁡dik⋅pkj#.

The improved drug-pathway matrix D~ was calculated by multiplying matrices D and P, where D→iT represented the ith row vector of matrix D and P→j represented the jth column vector of matrix P. In addition to assessing the effects of drugs on pathways, the D~ matrix also considered the interactions between pathways based on their similarities.

The final drug combination-pathway matrix A was obtained from the drug-pathway matrix D and D~:

(4) A=[a1,1,1⋮am,m,1a1,1,2⋮am,m,2…⋱…a1,1,n⋮am,m,n],ai,j,k=12(di,k⋅d~j,k+dj,k⋅d~i,k)#.

where ai,j,k represented the impact of the combined use of drug i and drug j on pathway k. In the end, the matrix A obtained through three-dimensional matrix operations not only considered the effects of drug combinations on pathways but also considered the mutual interactions between pathways.

Multi-omics data processing for cell lines

The multi-omics data describing cancer cell lines, including EXP (expression), MUT (mutations), CNV (copy number variations), METHY (methylation), and RNAi (RNAi-related data), were downloaded from the Cancer Cell Line Encyclopedia (CCLE) (Nusinow et al., 2020) and the Dependency Map (DepMap 22Q2) portal (http://depmap.org).

The level of gene expression reflected the level of gene activity in a cell and was a key link between functional genes and phenotypic changes. Gene expression data was obtained by RNA sequencing and processed using the RSEM tool to obtain Transcripts Per Million (TPM) expression values. We obtained 19,221 genes across 1,406 cell lines after a log2(TPM+1) transformation.

Gene mutations led to the loss or gain of protein function, directly affecting cell function and the occurrence and development of diseases, helping to identify key disease-causing genes and potential targets for targeted therapies. The mutation data was derived from the collective output of MuTect1, MuTect2, and Strelka, encompassing information on all somatic point mutations and indels called in DepMap. This dataset contained mutation information for 18,784 genes in 1,771 cell lines.

Copy number variation affected gene expression and function by altering gene dose, which might lead to tumor formation or the development of other complex diseases. Copy number variation scores were calculated by mapping genes to the segment-level calls and computing a weighted average along the genomic coordinate. There were copy number variation information for 25,368 genes in 1,766 cell lines after log2(CN ratio + 1) transformation.

DNA methylation referred to the addition of methyl groups to DNA molecules and was usually associated with the inhibition of gene expression. It played an important role in cell differentiation, development and disease. Methylation data was obtained by quantifying CpG islands using reduced representation bisulfite sequencing. Following Wang et al. (2020), we first removed genes that showed NA in more than 70% of cell lines. Subsequently, we utilized the K-nearest neighbor (KNN) to impute the remaining NA values. Finally, we calculated the average methylation value for each gene across all cell lines, resulting in data for 20,183 genes and 842 cell lines.

RNA interference mechanisms regulated gene expression by targeting specific mRNAs and was used to identify key genes that drove changes in cell function. RNAi data processing followed a similar approach to methylation data. Genes with NA values in more than 70% of the cell lines were first removed. Subsequently, the KNN was applied to impute the remaining NA values. The final dataset contained RNAi information for 12,500 genes and 710 cell lines after calculating the mean value.

There were 38 common cell lines between DepMap and NCI ALMANAC.

Cell line-pathway features

To provide a more comprehensive interpretation of the biological mechanisms of the model, we used pathways as a bridge connecting drug features with cell line features, which required the construction of a cell line-pathway matrix. For the four sets of cell line features (expression, copy number variation, methylation, and RNAi), we utilized the Gene Set Variation Analysis (GSVA) (Hanzelmann, Castelo & Guinney, 2013) method to obtain cell line scores for pathways. We employed the Jaccard coefficient to determine the overlap between genes with mutations and the gene set of pathways for each cell line. This process resulted in a mutation-related cell line-pathway feature matrix. Each set of cell line omics data results in 157 features.

NCI-ALMANAC data preprocessing

The NCI-ALMANAC dataset comprises 5,356 drug combinations, 104 drugs, 60 cell lines, and nine types of cancer. We first excluded all relevant data about single-drug actions. Then, since both NSC753082 and NSC761431 represent the drug vinorelbine, we removed the data related to NSC753082.

The drug combinations in NCI-ALMANAC were treated at different concentrations (3*3 or 5*3) in the same cell line (Xia et al., 2018). We averaged the synergy scores at different concentrations for each drug combination and cell line, named as CombScore.

(5) Combscore=1n∑a,b⁡Ea,b−Pa,b#.

We did not consider the concentration of drugs A and B. The mean difference between the expected growth score and the observed percent growth score was used as the CombScore.

We defined the drug pairs in each cell line as synergistic or not based on the CombScore. We categorized the CombScore using a threshold of 4 and compared it to the previous study using a threshold of 10 (Li et al., 2020). Drug pairs with CombScore greater than the threshold were considered synergistic, otherwise they were considered non-synergistic.

Drug combination modeling with machine learning and deep learning algorithms

We utilized five machine learning models—logistic regression, lasso regression, random forest, support vector machine (SVM) and eXtreme Gradient Boosting (XGBoost), along with one deep learning model, TabNet (Joseph, Joseph & Prasad, 2022). These models were trained on the pre-processed feature sets to evaluate their predictive performance on drug combination synergy.

Model adjustment and validation assessment

To evaluate the predictive performance of all machine learning models, we randomly divided the entire dataset into training and test sets with an 80:20 ratio. By using the StandardScaler from the scikit-learn library for feature normalization, we ensured that the features of both the training and test sets were adjusted to follow a standard normal distribution with a mean of 0 and a standard deviation of 1.We employed the GridSearchCV function from scikit-learn to perform a grid search and select the optimal parameters for improved performance. To address the issue of an imbalanced sample distribution, we employed techniques such as adjusting sample prediction weights, undersampling, and oversampling during the model development process, aiming to rebalance the ratio of negative to positive samples to 2:1. The sampled dataset was further divided into five equal-sized subsets for five-fold cross-validation to avoid problems such as oversampling after data processing and to ensure the stability of model predictions. Evaluation metrics including accuracy, precision, recall, F1-score, area under the receiver operating characteristic curve (AUC), and area under the precision-recall curve (AUPR) were used to evalute the binary classification models. Due to the imbalance between negative and positive samples, we focused primarily on AUPR values to assess the predictive ability of the models, emphasizing the correct prediction of true positive samples.

Drug validation experiments

We cultured T-47D cells (human mammary ductal carcinoma cells) purchased from Pricella and used them within 2 weeks of receipt. The cells were characterized by short tandem repeat (STR) profiling. The cell lines were passaged until growth reached 70–85% of the flask. The passaging process involved washing the residual culture medium with phosphate-buffered saline (PBS), digesting the cells with tryptic cell digest, and terminating the digestion with T-47D cell-specific medium (Pricella, CM-0228), which was prepared as a cell suspension. Passaging was performed every 48 h at a ratio of 1:3. During cell processing, fresh culture medium was prepared and cell counting was performed by diluting the cell suspension to 50,000 cells/ml. The single-cell suspension was then spread and 200 μL of single-cell suspension was added per well in a 96-well plate, resulting in 10,000 cells per well. The cells were incubated in a cell culture incubator at 37 °C, 5% CO2, and 95% humidity for 24 h (Pettersen et al., 2007).

For drug treatments, all drugs were purchased from Selleck. For single drug studies, cells were treated with six concentrations of bortezomib (0.5, 2.5, 10, 20, 30, and 40 nmol/L), with each concentration repeated eight times. In addition, cells were treated with four concentrations of bleomycin (50, 100, 200, and 400 nmol/L), and the experiment was also repeated eight times for each drug concentration. Cell survival was determined after 48 h and the optimal concentration was selected based on the rate of inhibition of cell growth by each drug. To study drug combinations, eight replicate treatments were performed with two drugs added to the cell lines at the optimal concentration, and the experiment was repeated twice to account for the order of drug administration. The results of the two experiments were recorded as Combination 1 and Combination 2.

Finally, cell viability assays were performed using the CCK-8 method by aspirating the drug-containing medium and adding 100 μL of fresh medium (Cai et al., 2019). To each well, 10 μL of CCK8 activity assay reagent was added and incubated for 1 h at 37 °C, 5% CO2 and 95% humidity. Cell activity was detected using a microplate reader at 450 nm. The cell activity formula was calculated as follows:

(6) Cellviability(%)=[Ad/An]×100%#.

We compared the effectiveness of single drugs and drug combinations using the method of Li et al. (2020):

(7) IDAcombscore=△via−△via×HRC/Mbest#

where △via was equal to the mean viability when cell lines were treated with the best monotherapy minus the mean viability when cell lines were treated with the drug combination. HRC/Mbest was the hazard ratio between the drug combination efficacy and the best monotherapy efficacy.

Result

Exploration of NCI-ALMANAC dataset

The NCI-ALMANAC dataset was one of the largest known drug combination datasets, containing 104 drugs, 5,356 drug pairs and 60 cell lines across nine cancer types, with each cancer type containing 2–9 cell lines. To mitigate the impact of data sparsity on the input features, using a Combscore threshold of four as an example, we selected 87 drugs with at least 10 targets each, resulting in 3,741 drug pairs, some of which have been experimentally validated by the NCI (see Figs. 2A, 2B). When the threshold is set to 10, specific details of drug combinations are illustrated in Fig. S1. We selected 38 cell lines with comprehensive information on expression, mutation, copy number variation, methylation, and RNAi (Figs. 2C, 2D). In total, 128,445 drug-drug-cell line data points were retained for model training, with an additional 13,713 data points reserved for testing (see Table S1). Simultaneously, we downloaded 340 human-relevant pathways from KEGG, filtering for those in which at least half of the drugs acted and where each pathway contained at least 10 genes. This resulted in a final set of 157 pathways.

Figure 2 Exploration and filtering of the NCI-ALMANAC dataset.

(A) The stacked bar chart illustrates the number of drug pairs in different cell lines, taking the threshold of 4 as an example. Green represents the number of drug pairs without synergistic effects according to NCI, yellow represents the number of drug pairs with synergistic effects according to NCI, and red represents the number of untested drug pairs in the validation set by NCI. (B) Positive drug pair counts in cell lines corresponding to the nine types of cancer when the threshold is set to 4. (C) The Venn diagram illustrates the intersection between the five cell line omics data and the NCI60 cell lines. The UpSet plot displays the number of unique cell lines in each combination. (D) The bar chart above illustrates the number of intersecting cell lines in each combination, with marked points below indicating the involved omics types in the combinations.

Due to the limited number of positive samples, the sparsity of the input feature matrix may affect the accuracy of predicting synergistic effects. Therefore, we applied specific criteria to filter drugs, pathways, and cell lines (see Methods for details). The feature representation of each sample is a 1,148-dimensional vector. After normalizing the input features, we employed various models for training, including linear regression, lasso regression, random forest, support vector machine, XGBoost and TabNet, and comparing their results.

Model selection and evaluation

To compare the model performance under two different thresholds, we used all feature sets (1,148 features) as inputs for model training. With a threshold of 4, there are 14,371 positive samples (drug-drug-cell line combinations with synergistic effects) and 114,075 negative samples. With a threshold of 10, there are 3,575 positive samples and 124,871 negative samples. Due to the imbalanced distribution of the classification results in the samples, we adopted weight assignment to increase the probability of predicting positive samples, with the default being ‘class_weight = balance’.

Comparing the results of the six models under the two thresholds (Table 1), it could be observed that the models with a threshold of 4 had higher precision, recall and AUPR values (Fig. 3A), whereas the models with a threshold of 10 had higher accuracy and AUC values (Fig. 3B). Considering the imbalance in the distribution of positive and negative samples, we finally chose the threshold with higher AUPR values, i.e., a threshold of 4, as the criterion for determining whether drug combinations show synergistic effects.

Table 1 The comparative evaluation of models under two thresholds.

Model	Threshold = 4	Threshold = 10	
ACC	PREC	Recall	F1 Score	AUC	AUPR	ACC	PREC	Recall	F1 Score	AUC	AUPR	
Logistics	0.891	0.546	0.071	0.125	0.760	0.319	0.972	0.568	0.035	0.066	0.846	0.213	
Lasso	0.890	0.544	0.068	0.121	0.758	0.314	0.972	0.558	0.034	0.063	0.844	0.211	
RF	0.896	0.866	0.071	0.131	0.810	0.468	0.973	0.738	0.067	0.123	0.862	0.293	
SVM	0.910	0.720	0.310	0.433	0.832	0.531	0.975	0.616	0.212	0.347	0.870	0.383	
XGBoost	0.911	0.705	0.345	0.463	0.865	0.564	0.975	0.665	0.228	0.34	0.909	0.424	
TabNet	0.904	0.642	0.300	0.409	0.827	0.476	0.974	0.577	0.178	0.272	0.865	0.296	

Figure 3 Two thresholds evaluation of machine learning models.

(A) The bar chart illustrates the evaluation metrics for the five models under the threshold of 4. (B) The bar chart illustrates the evaluation metrics for the five models under the threshold of 10.

By comparing the various evaluation metrics of the six models, the overall performance of the XGBoost model outperformed the other models. The performance and training time of XGBoost is significantly influenced by the choice of parameters. To improve the accuracy of the model predictions, we used grid search to determine the values of several commonly used variables in XGBoost: ‘learning_rate’ was set to 0.1, ‘max_depth’ was set to 7, and ‘n_estimators’ was set to 700. In addition, we utilized a five-fold cross-validation method to prevent over- or under-fitting of the models on certain datasets, with the best AUC and AUPR being 0.865 and 0.564 respectively. We continued to use lattice search and cross-validation in subsequent model adjustments to improve the predictive performance of the models.

Feature selection

We tested our model on different combinations of feature categories to select the optimal input feature set. There were three types of drug features, including MACCS molecular descriptors, drug toxicity, and pathways, and five types of cell line features, including expression, copy number variation, mutation, RNAi, and methylation. We randomly combined these eight feature sets into a total of 255 input feature sets. Subsequently, we utilized XGBoost with the same optimal parameters for modeling and evaluated the results using the AUPR value (details in Table S2).

The evaluation results, displaying the top 5 AUPR values and the remaining combinations of multiple feature sets, were presented in Table 2. Considering that recall, F1-score, and AUPR were commonly used metrics to evaluate imbalanced data classification, we ranked these three metrics separately and calculated their average ranks. The table shows that the model using a combination of seven feature sets—MACCS, toxicity, pathway, expression, CNV, RNAi, and methylation—performed the best, achieving an AUPR value of 0.5604 and ranking first. It can also be observed from Figs. 4A–4C that it demonstrated a high F1 score and Recall values. On the other hand, the model using all eight feature sets ranked only eighth in terms of AUPR.

Table 2 The evaluation metrics and overall rankings for selected combinations of feature sets.

Feature combination	ACC	PREC	Recall	F1 Score	AUC	AUPR	Average rate	
DD_MACCS, DD_TOX, DDP, EXP, CNV, METHY, RNAi	0.912	0.706	0.354	0.472	0.864	0.560	1	
DD_MACCS, DD_TOX, DDP, EXP, CNV, MUT	0.912	0.705	0.355	0.472	0.865	0.559	2	
DD_MACCS, DDP, EXP, CNV, MUT	0.911	0.698	0.353	0.469	0.863	0.560	3	
DD_MACCS, DD_TOX, DDP, EXP, CNV, METHY, MUT	0.911	0.696	0.353	0.469	0.864	0.560	4	
DD_MACCS, DDP, EXP, METHY	0.911	0.694	0.355	0.47	0.864	0.558	5	
DD_MACCS, DD_TOX, DDP, EXP, CNV, METHY, MUT, RNAi	0.91	0.69	0.347	0.462	0.864	0.559	26	
DD_TOX, DDP, CNV, MUT, RNAi	0.909	0.685	0.326	0.441	0.858	0.539	136	
DD_TOX, DDP, CNV, MUT	0.909	0.689	0.325	0.442	0.857	0.538	137	
DD_TOX, DDP, CNV	0.909	0.69	0.326	0.443	0.856	0.535	140	
DD_MACCS	0.905	0.66	0.293	0.405	0.833	0.502	159	
DD_TOX, DDP	0.905	0.661	0.293	0.406	0.834	0.502	160	
DDP, CNV, METHY, MUT	0.905	0.667	0.292	0.406	0.838	0.496	161	
DDP, METHY	0.906	0.680	0.288	0.404	0.837	0.494	166	
DDP, CNV	0.905	0.666	0.29	0.404	0.838	0.491	170	
DDP	0.903	0.655	0.259	0.371	0.82	0.47	192	
DD_TOX	0.901	0.692	0.199	0.309	0.804	0.448	204	
EXP	0.889	0	0	0	0.576	0.16	225	
CNV	0.889	0	0	0	0.576	0.16	226	
METHY	0.889	0	0	0	0.576	0.16	227	
MUT	0.889	0	0	0	0.576	0.16	228	
RNAi	0.889	0	0	0	0.576	0.16	229	

Figure 4 Evaluation metrics for all feature combinations.

(A) Recall, (B) F1 score, and (C) AUPR for all feature set combinations. The blue represents the final selected seven feature set combinations, while the red represents combinations using all feature sets.

Meanwhile, we observed that using a single feature set, whether it’s drug related or cell line related, the predictive performance is in the bottom 50% across all models. Specifically, when using only cell line features, the precision, recall, and F1 values all equate to 0. This may be attributed to the small sample size of 38 cell lines compared to the total samples of 14,371 positives and 114,075 negatives, causing the predictions to heavily favor negative samples, indicating no synergistic effect of drug combinations on cell lines. However, when combining drug and cell-line features, the predictive performance of most models improves. Taking drug pathway features as an example, the inclusion of methylation data increases the ranking of the model by 21 positions compared to using cell line pathways alone. In addition, using multiple cell line data is more effective than adding a single cell line data. For example, modeling with only drug pathways and methylation data performs less effectively than models incorporating drug pathways along with CNV, methylation, and mutation data. Furthermore, we observed that using only drug features is less effective than combining drug and cell line feature sets. Taking drug toxicity and drug pathways as an example, their predictive performance is inferior to the feature set that includes CNV data. In addition, introducing more cell line features slightly improves the predictive performance of the model.

In summary, we ultimately chose the feature set consisting of MACCS, drug toxicity, drug pathways, gene expression, copy number variation, RNAi, and methylation as the input features for the model. With this feature set, we achieved the best evaluation results using XGBoost.

Correction of unbalanced data

To address the issue of imbalanced sample distribution, we employed the following three methods to re-model the previously selected optimal feature set combination. (R1) Adjusting positive and negative sample weights: Given the larger proportion of negative samples, we adjusted the weights based on the ratio of negative to positive sample quantities. This adjustment aids convergence in the context of imbalanced samples. (R2) Undersampling: Using the imblearn package from sklearn, we reduced the number of positive samples to achieve a balanced ratio of 1:2 between positive and negative samples. (R3) Oversampling: Employing the SMOTE method, we synthesized samples for the minority class, maintaining a balanced ratio of 1:2 between positive and negative samples. The evaluation results of establishing models using the three methods are shown in Table 3.

Table 3 Evaluation results among different imbalanced data handling model.

Method	Model	ACC	PREC	Recall	F1 Score	AUC	AUPR	
R1: adjusting weights	Logistics	0.8905	0.5462	0.0706	0.1251	0.7604	0.3194	
Lasso	0.8904	0.5437	0.0678	0.1206	0.7581	0.3137	
RF	0.8958	0.8664	0.0706	0.1306	0.8104	0.4685	
SVM	0.9102	0.7204	0.3096	0.4330	0.8323	0.5319	
XGBoost	0.9114	0.7050	0.3450	0.4633	0.8654	0.5638	
TabNet	0.9039	0.6418	0.2997	0.4086	0.8273	0.4755	
R2: undersampling	Logistics	0.7381	0.6604	0.4684	0.5480	0.7730	0.6334	
Lasso	0.7400	0.6654	0.4687	0.5500	0.7716	0.6318	
RF	0.7676	0.7925	0.4259	0.5541	0.8109	0.7186	
SVM	0.8020	0.7527	0.6196	0.6797	0.8352	0.7445	
XGBoost	0.8177	0.7886	0.6315	0.7014	0.8688	0.8054	
TabNet	0.7960	0.7447	0.6059	0.6682	0.8456	0.7621	
R3: oversampling	Logistics	0.7483	0.6562	0.5006	0.5679	0.7959	0.6392	
Lasso	0.7474	0.6559	0.4951	0.5643	0.7941	0.6375	
RF	0.8912	0.9455	0.7118	0.88122	0.9495	0.9262	
SVM	0.9169	0.8924	0.8511	0.8713	0.9674	0.9453	
XGBoost	0.9341	0.9630	0.8325	0.8930	0.9685	0.9595	
TabNet	0.8616	0.8814	0.6715	0.7623	0.9187	0.8835	

Among the three methods, the oversampling approach consistently outperformed the weighting and undersampling approaches in terms of various evaluation metrics. Although the method of weight allocation can yield a better ACC value, but its prediction results tended to favor negative samples, which was contrary to our goal of obtaining more positive samples. By referring to Figs. 5A, 5B, in the ROC and PR curves of AUC and AUPR, the areas under the curves for the XGBoost model after oversampling processing reached 0.9685 and 0.9595, respectively, demonstrating excellent predictive performance. The ROC and PR curves for the other two methods were shown in Fig. S2.

Figure 5 Evaluation of results after oversampling method handling.

Panels (A) & (B) respectively depict the ROC curves and PR curves of each model after oversampling, representing their AUC and AUPR values. Panels (C)–(F) represent the recall, F1-score, AUC, and AUPR values of each model after the three different methods, where R1 represents weight adjustment, R2 represents undersampling, and R3 represents oversampling.

In addition, as our focus was more on the predictive performance of positive samples, we aimed to achieve better recall, F1-score and AUPR during the model evaluation process. Through a horizontal comparison of multiple models under the same method, the results (see Figs. 5C–5F) indicated that regardless of the method used to handle imbalanced data, the XGBoost model exceled in the classification task of predicting synergistic effects. Compared to other classical machine learning models commonly used for such tasks, such as logistic regression, lasso, SVM, and random forest, the XGBoost model showed a significant improvement in various evaluation metrics, including notable enhancements in accuracy and precision (see Figs. S3E–S3F). Therefore, we ultimately decided to adopt the XGBoost model trained with oversampling as the preferred training model.

Feature importance evaluation

To better explain the biological mechanisms behind drug synergies, we connected information on drug combination effects to cell line information via biological pathways. We selected five feature sets related to pathways from the seven feature sets and calculated the importance of each feature (details in Table S3). These included one drug feature (drug pair pathway) and four cell line features (expression-pathway, methylation-pathway, CNV-pathway, and RNAi-pathway). From the A, it could be observed that the contributions of these five feature sets were relatively similar, accounting for 47% of the total feature importance. Despite the drug MACCS fingerprint being the most important feature set, acknowledged as an excellent feature for studying drug synergies, it lacked relevant support for understanding the biological mechanisms of drug synergy in our research.

To investigate the most important pathway information among the five feature sets, we performed a more detailed analysis. We selected the top one, two, three, four, five, and 10 features from each feature set, repeated the modeling using the XGBoost model, and plotted the trends of various evaluation metrics (Fig. 6B). We found that using the top 10 features in each feature set, ranked by contribution, as input data can essentially achieve the same results as training with all features.

Figure 6 Feature importance analysis.

(A)The proportion of feature importance for each feature set. (B) Line charts depicting the model evaluation metrics for the top one, two, three, four, five, and 10 features selected from each feature set, as well as for all features. (C) Functional annotation of the top ten pathway features in each feature set ranking.

Summarizing the top ten features (pathways) from each feature set (Fig. 6C), we observed that the pathways associated with each feature set were almost unique. Only the linoleic acid metabolism pathway ranked in the top ten for both CNV and RNAi features, and the prion disease pathway was prominent in METHY and RNAi. To understand the functional information associated with these pathways, we annotated them based on the primary functions in KEGG. Most of the top ten pathways for each feature set were related to human diseases functions. However, there were still distinctions in emphasis among feature sets. For instance, DDP still includes pathways related to environmental information processing, whereas CNV features were associated with organismal systems. In particular, among the top three ranked features (Table S4), DDP pathways were mainly related to environmental information processing, EXP was predominantly linked to organismal systems, CNV, and METHY primarily involved pathways related to human diseases, and RNAi showed a broader range of pathways. This also indicated that different omics information provided distinct biological pathway insights, enhancing the mechanistic scope of biological pathways through multi-omics information.

Prediction and validation in unlabeled data

In NCI-ALMANAC, only partial experimental data on the interaction of drug combinations with cell lines are provided. We processed the remaining drug combination information in the same way, including 113,713 drug combination-cell line data as a validation set. We used the trained model to make predictions, and the predicted results were shown in Table S5. Among these, there were 287 positive results, indicating drug combinations predicted to have a synergistic effect on cell lines. Figures 7A–7D illustrated all synergistic drug combination-cell line pairs predicted for breast cancer, leukemia, non-small cell lung cancer, and ovarian cancer, respectively. Results for other cancers are shown in the Fig. S3.

Figure 7 Predicted results for synergistic drug combinations.

(A) The predicted drug combinations that can act on breast cancer-related cell lines are shown in the results. The red nodes represent drug combinations, the green nodes represent cell lines, and the blue nodes represent cancer types. Nodes with yellow outer circles indicate results validated through drug experiments. (B) The predicted drug combinations that can act on ovarian cancer-related cell lines are shown in the results. Nodes with green outer circles indicate results validated through literature research. (C) The predicted drug combinations that can act on leukemia-related cell lines are shown in the results. (D) The predicted drug combinations that can act on non-small cell lung cancer-related cell lines are shown in the results.

To validate the authenticity of the predicted results, we conducted a literature search on all positive outcomes and selected one for experimental verification. Three cases were validated through the literature. Specifically, the combination of doxorubicin hydrochloride and celecoxib showed a synergistic effect on the HL-60 (TB) cell line. The combination of 4′-epiadriamycin and erlotinib hydrochloride inhibits the growth of the A549 cell line, and Arsenic trioxide in combination with cisplatin acts on the ovarian cancer-related IGROV1 cell line.

He et al. (2015) research indicated that in HL-60 cell lines, both MDR1 and COX2 mRNA were both expressed at low levels. After treatment with 0.02 μg/ml daunorubicin for 72 h, the expression of MDR1 and COX2 mRNA increased by 3.68 and 3.29 times, respectively. Subsequent treatment with celecoxib revealed that the induction of MDR1 mRNA expression by daunorubicin was significantly upregulated, while COX2 mRNA showed a slight decrease. Overall, compared to daunorubicin alone, the combination with celecoxib resulted in a 3.07-fold decrease in the effective inhibitory concentration (IC50) for HL-60 compared to daunorubicin alone, and the use of celecoxib alone did not inhibit the growth of HL-60 cell lines.

Similar to the combination of daunorubicin and celecoxib, He et al. (2016) suggested that the combination of daunorubicin and erlotinib can inhibit the growth of A549 cell lines associated with non-small cell lung cancer. Additionally, research by Sweatman & Israel (1987) indicates that adriamycin (ADR) and 4′-epiadriamycin (epi-ADR) showed comparable biological properties in most in vitro and in vivo test systems. Moreover, 4′-epiadriamycin has lower cardiac toxicity. Therefore, we infer that the combination of 4′-epiadriamycin and erlotinib can also inhibit the growth of A549 cell lines. Additionally, Zhang et al. (2009) found that compared to using cisplatin or arsenic trioxide alone, the combination of cisplatin and arsenic trioxide can inhibit the growth of ovarian cancer-associated IGROV1 cell lines at lower doses. The dose reduction index (DRI) ranges from 1.75 to 5.21 under different concentrations.

Drug validation experiments

At the same time, we carried out drug experiments on the synergistic inhibitory effect of bleomycin and bortezomib on breast cancer cell line Tmur47D. We measured the cell proliferation activity of bleomycin at four concentrations and bortezomib at six concentrations for 48 h and determined that the optimal concentration of bleomycin was 200 nmol, and that of bortezomib was 40 nmol/L. The results of the drug experiments in Table 4 showed the effect of the combination of two drugs on cell activity at the optimal concentration through multiple repeated experiments. The combination experiments for the two drugs were conducted repeatedly, employing the studied drug combination efficacy calculation method of Ling & Huang (2020). Using the cell activity detected for each single drug and the cell activity of the combination drug, the obtained IDAcomboscore values were 0.014 and 0.018, respectively, both exceeding 0.004 (see Fig. 7E). Therefore, these two drugs were considered to have synergistic effects.

Table 4 The experimental results of drug treatment at the optimal concentration.

Group	Bleomycin	Bortezomib	Combination 1	Combination 2	
1	0.8651	0.7793	0.6603	0.6315	
2	0.8308	0.7798	0.7207	0.5755	
3	1.1593	0.8857	0.7435	0.6951	
4	0.8374	0.8288	0.7579	0.7582	
5	0.8043	0.8641	0.5590	0.6176	
6	0.7598	0.7557	0.6598	0.6719	
7	0.6332	0.5992	0.5037	0.4989	
8	0.8417	0.6336	0.6431	0.6852	

Discussion

Chemotherapy was one of the traditional treatment methods for early-stage cancer. However, the efficacy of tumor chemotherapy was often limited by drug resistance, drug toxicity, and tumor heterogeneity. With the continuous development of pharmacology, oncology, and clinical trial technologies, utilizing synergistic drug combinations in combination chemotherapy provided a new approach to addressing these issues. That was, combining two drugs with similar targets may enhance efficacy and reduce side effects through interrelated biological processes, thus offering a promising solution to these challenges.

In this study, we developed the XDDC model to predict the synergistic effects of drug combinations, utilizing drug pathway information to connect drug features with cell line characteristics. Through modeling and evaluating different combinations of feature sets, we found that the best predictive performance was achieved when using drug structure, drug toxicity, pathway interactions, gene expression, methylation, copy number variation, and RNA interference data as input feature sets. Furthermore, by comparing different methods for handling imbalanced data, we observed that using oversampling significantly improved the model’s AUC and AUPR values compared to the other two methods. Subsequently, by annotating the top ten pathways in pathway-related feature sets, we discovered that different omics information provided distinct biological pathway insights during the prediction of drug combinations.

Finally, we utilized the XDDC model to predict the test set of NCI-ALMANAC and obtained 287 positive results. Among these, we selected a subset of positive results for experimental validation. The experimental results confirmed that the combination of bleomycin sulfate and bortezomib synergistically inhibited the T-47D breast cancer cell line, with the combined treatment showing significantly better efficacy than monotherapy. The XDDC model can help to rapidly determine the efficacy of drug combinations, and it is expected to be a powerful tool in the process of drug development and combination therapy. However, there are some limitations to this study. The data on drug combination synergy in cell lines is still limited, which makes it difficult to construct a model with high accuracy rates. The imbalance of the positive and negative samples makes it more complex, although we have tried some sampling methods to prevent fitting problems. Due to technical and ethical constraints, we have only tested a set of drug combinations and cell lines whose effects have not been tested in clinical trials. Even if the models, machine learning and deep learning, can make accurate predictions, the internal working mechanism is often opaque. It makes it difficult to explain the model features from a biological point of view, so we can only provide some relatively superficial explanation based on the importance of the features. We believe that with the completion of more experimental data on drug combinations, the expansion of the size of the training set for synergistic drug combinations, and the refinement of the properties of drugs, the XDDC model will help to rapidly determine the efficacy of drug combinations and is expected to become a powerful tool in drug development and combination therapy, based on the known effects of the drugs, and borrowing the means of computers and experiments to deduce and preliminarily screen drug combinations for use in combination. The use of computational and experimental tools to infer and preliminarily screen drug combinations based on known drug effects will lead to more effective and economical therapeutic benefits for patients.

Supplemental Information

Supplemental Information 1 Exploration and filtering of the NCI-ALMANAC dataset.

(A) The stacked bar chart illustrates the number of drug pairs in different cell lines, taking the threshold of 10 as an example. Green represents the number of drug pairs without synergistic effects according to NCI, yellow represents the number of drug pairs with synergistic effects according to NCI, and red represents the number of untested drug pairs in the validation set by NCI. (B) Positive drug pair counts in cell lines corresponding to the 9 types of cancer when the threshold is set to 10.

Supplemental Information 2 Weight allocation and undersampling method evaluation.

A-B represents the ROC curves of the models after weight adjustment and undersampling, indicating the AUC values. C-D represents the PR curves of the models after processing, indicating the AUPR values. E-F represent the accuracy and precision of each model after the three different methods, where R1 represents weight adjustment, R2 represents undersampling, and R3 represents oversampling.

Supplemental Information 3 Prediction of drug combination networks.

A-E represent the predicted drug combination networks that can act on melanoma, central nervous system cancer(CNS cancer), renal cancer, colon cancer, and prostate cancer-related cell lines.

Supplemental Information 4 Overview of NCI-ALMANAC Training and Testing Sets.

Supplemental Information 5 Evaluation of Combined Input Feature Sets.

Supplemental Information 6 Feature Importance.

Supplemental Information 7 The top ten featured pathways and their annotations in each feature set.

Supplemental Information 8 All positive predicted results.

Additional Information and Declarations

Competing Interests

The authors declare that they have no competing interests.

Author Contributions

Jiaqi Chen conceived and designed the experiments, performed the experiments, analyzed the data, prepared figures and/or tables, authored or reviewed drafts of the article, and approved the final draft.

Huirui Han analyzed the data, prepared figures and/or tables, investigation, Methodology, and approved the final draft.

Lingxu Li performed the experiments, prepared figures and/or tables, and approved the final draft.

Zhengxin Chen conceived and designed the experiments, performed the experiments, prepared figures and/or tables, conceptualization, Investigation, and approved the final draft.

Xinying Liu performed the experiments, authored or reviewed drafts of the article, data curation, Visualization, and approved the final draft.

Tianyi Li analyzed the data, prepared figures and/or tables, methodology, Resources, and approved the final draft.

Xuefeng Wang analyzed the data, authored or reviewed drafts of the article, data curation, Formal analysis, and approved the final draft.

Qibin Wang performed the experiments, authored or reviewed drafts of the article, and approved the final draft.

Ruijie Zhang analyzed the data, authored or reviewed drafts of the article, resources, Validation, and approved the final draft.

Dehua Feng analyzed the data, authored or reviewed drafts of the article, formal analysis, Methodology, and approved the final draft.

Lei Yu performed the experiments, prepared figures and/or tables, formal analysis, Methodology, and approved the final draft.

Xia Li conceived and designed the experiments, authored or reviewed drafts of the article, methodology, Project administration, Writing – review & editing, and approved the final draft.

Limei Wang conceived and designed the experiments, authored or reviewed drafts of the article, methodology, Resources, Validation, Writing – review & editing, and approved the final draft.

Bing Li conceived and designed the experiments, performed the experiments, authored or reviewed drafts of the article, supervision, Validation, Writing – review & editing, and approved the final draft.

Jin Li conceived and designed the experiments, performed the experiments, analyzed the data, prepared figures and/or tables, authored or reviewed drafts of the article, funding acquisition, Project administration, Validation, Visualization, Writing – review & editing, and approved the final draft.

Data Availability

The following information was supplied regarding data availability:

Data and code are available at Zenodo:

Chen, J. (2024). Prediction of cancer cell line-specific synergistic drug combinations based on multi-omics data [Data set]. Zenodo. https://doi.org/10.5281/zenodo.14552135.

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
