# Peer review of "Prediction of cancer cell line-specific synergistic drug combinations based on multi-omics data"

_PeerJ, doi:10.7717/peerj.19078_

## Round 0.1 · original submission · Major Revisions

Thank you for your submission. While positive, the reviewers bring up a number of important questions and concerns.

Reviewer 1 points out that features should be explained more clearly and that a broader discussion of the strengths and weaknessess of tis approach would be valuable.

Reviewer 2 in particular brings up a number of valid points regarding the literature background provided as well as the broader approach. The response prediction variable(s) should be defined in more detail and it should be noted how this approach and its strengths and weaknesses compare to other methods. The comment by reviewer 1 is pertinent here as the synergistic/not synergistic approach used in this work is not what is typically standard in the field, and comparisons to other methods are difficult as a result. Establishing predictions that are potentially more comparable, e.g., for instance to TabNet would be valuable. For broader reproducibility, full description of the data along with relevant code should be provided, ideally in a repository such as github.

Reviewer 1 ·

Basic reporting

no comment

Experimental design

no comment

Validity of the findings

no comment

Additional comments

There are some minor issues:
1. English should be improved to fix some typos and grammar issues. For example, in line 87, "popular" should be "popularly" or "widely"; line 127, "This dataset includes" is an uncompleted sentence.

2. The fonts in the figures are too small for authors.

3. The biological explanation of the selected features is too limitted. For example, why is DNA methylation a selected feature? What is it indicated?

4. The Discuss part did not discuss the limitations of their method, which should be very useful for further improvement.

Reviewer 2 ·

Basic reporting

The study describes an innovative effort to create a modelling scheme for drug combination efficacy prediction that relies on pathway dependencies of drug targets and their hypothetical effects on the pathways. The literature search was fairly thorough, but none of the papers cited utilized drug-target data similar to how the authors use it. I recommend adding some citations for models that include drug-target data such as 10.1093/bib/bbad034, 10.1186/1752-0509-8-74.

Experimental design

The mathematical basis for the drug-target and cell-line dependent pathway model is clearly explained and well thought out. However, the response prediction variable is not clearly stated, although it seems to be the synergy score. The choice of a classification model to predict a synergy score (Comboscore) is unusual, since almost all of the cited prior models from the literature predict precise cell viability and synergy scores using regression models instead of classifying simply as "synergistic" or "not synergistic". This level of prediction seems far too simplistic compared to what is currently in the literature. Additionally, machine learning models should be tested against the most current algorithms built for tabular data, such as TabNet or a similar transformer/deep neural network. Overall, I would recommend predicting precise cell viability and synergy scores, producing R-squared values, and comparing the model performance to previous models from the literature that have trained on the same data (NCI-Almanac).

Validity of the findings

The experimental validation is well thought out and helps with validating the described model architecture. However, the validity of the findings is in question as none of the training data, results, testing data, predictions, or code used for the findings seem to be publicly available on any platform. I believe this is necessary for any computational effort.

---

## Round 0.2 · accepted · Accept

Thank you for addressing the reviewers' concerns and congratulations again.

Reviewer 1 ·

Basic reporting

no comment

Experimental design

no comment

Validity of the findings

no comment

Additional comments

no comment

Reviewer 2 ·

Basic reporting

The authors have responded to feedback regarding the methods and content to a suitable degree, along with adding more context to the text that will help the readers understand the scope of the work.

Experimental design

The authors have made verifiable changes to the work based on the initial review

Validity of the findings

The authors have made their code and data available on github to enhance the validity of the findings.